

# Spatial structure of unstable normal modes
# in a glass-forming liquid

**Masanari Shimada[1*], Daniele Coslovich[2,3], Hideyuki Mizuno[1] and Atsushi Ikeda[1]**

**1** Graduate School of Arts and Sciences, The University of Tokyo, Tokyo 153-8902, Japan
**2** Laboratoire Charles Coulomb (L2C), Université de Montpellier, CNRS, Montpellier, France
**3** Dipartimento di Fisica, Università di Trieste, Strada Costiera 11, 34151, Trieste, Italy

⋆ masanari-shimada444@g.ecc.u-tokyo.ac.jp

## Abstract

The phenomenology of glass-forming liquids is often described in terms of their underlying, high-dimensional potential energy surface. In particular, the statistics of stationary points sampled as a function of temperature provides useful insight into the thermodynamics and dynamics of the system. To make contact with the real space physics, however, analysis of the spatial structure of the normal modes is required. In this work, we numerically study the potential energy surface of a glass-forming ternary mixture. Starting from liquid configurations equilibrated over a broad range of temperatures using a swap Monte Carlo method, we locate the nearby stationary points and investigate the spatial architecture and the energetics of the associated unstable modes. Through this spatially-resolved analysis, originally developed to study local minima, we corroborate recent evidence that the nature of the unstable modes changes from delocalized to localized around the mode-coupling temperature. We find that the displacement amplitudes of the delocalized modes have a slowly decaying far field, whereas the localized modes consist of a core with large displacements and a rapidly decaying far field. The fractal dimension of unstable modes around the mobility edge is equal to 1, consistent with the scaling of the participation ratio. Finally, we find that around and below the mode-coupling temperature the unstable modes are localized around structural defects, characterized by a disordered local structure markedly different from the liquid's locally favored structure. These defects are similar to those associated to quasi-localized vibrations in local minima and are good candidates to predict the emergence of localized excitations at low temperature.



# 1 Introduction

Glass formation is driven by the rapid increase of structural relaxation times that takes place when liquids are cooled fast enough to avoid crystallization [1,2]. From a purely operational point of view, the glass transition occurs at a temperature, called $T_g$, at which the structural relaxation time reaches about 100s. Below $T_g$, the supercooled liquid behaves, on typical experimental time scales, as an amorphous solid. However, the lack of a sharp divergence in dynamic and thermodynamic quantities, as would be observed in conventional phase transitions, makes it difficult to define the glassy state in a clear-cut way, and the mechanisms responsible for the enormous slowing down of the dynamics are still highly debated [3–5].

A long-standing line of thought describes glass formation in terms of the exploration of a high-dimensional potential energy surface (PES) [6,7]. Stationary points of the PES are configurations such that the gradient of the total potential energy $\mathcal{U}$ vanishes, and correspond to either local minima or saddles. These configurations play an important role in the PES-based description of glass formation. In his seminal work in 1969, Goldstein [8] argued that the dynamics of supercooled liquids is dominated by activated barrier crossing between neighboring local minima below some temperature $T_x$, at which the structural relaxation time is of order $10^{-9}s$. In this scenario, $T_x$ marks the crossover between the normal liquid dynamics and activated glassy dynamics over large energy barriers.

This phenomenological description later evolved into a more complex picture, which suggests the existence of an additional dynamic regime [7]. The early stages of the slowing down of several supercooled liquids are indeed reasonably well described by mode-coupling theory (MCT) [9]. MCT provides a semi-quantitative description of several non-trivial dynamic features above the critical temperature $T_{\mathrm{MCT}}$, at which the theory predicts a power law diver-

gence of the relaxation times. This singularity is smeared out in actual supercooled liquids by the presence of thermally activated processes, which are not accounted for by the theory and turn the transition into a crossover. According to a series of theoretical studies [10,11], however, a sharp change in the PES still underlies the MCT crossover: above $T_{\mathrm{MCT}}$, the dynamics is governed by the motion along unstable saddle modes, which the system can exploit without resorting to thermal activation; below $T_{\mathrm{MCT}}$, instead, the system mostly vibrates around local minima, occasionally relaxing via activated barrier crossing, and one can thus identify $T_{\mathrm{MCT}} \approx T_x$. This theoretical scenario is inspired by the behavior of a class of mean-field spin models, for which the (schematic) MCT equations are exact and the nearest stationary point to an equilibrium configuration indeed changes from a saddle to a minimum at the MCT transition [12–14].

Finding a clear signature of the MCT crossover in the PES of actual supercooled liquids is quite challenging and this has led to contradictory claims [15–25]. These difficulties were due to two main factors. First, conventional simulations cannot easily sample the PES at equilibrium below $T_{\mathrm{MCT}}$, due to exceedingly long equilibration times. Second, all the above mentioned studies neglected the role of the modes' spatial localization. In particular, it is clear that thermally activated processes in finite dimensional liquids involve only localized portions of the system, whereas the unstable modes of saddles in mean-field models are spatially delocalized. A recent numerical study [26], based on an efficient swap Monte Carlo algorithm [27,28], has tackled these two issues at once, establishing that the stationary points of several three-dimensional model liquids indeed show a sharp change around $T_{\mathrm{MCT}}$: the fraction of delocalized unstable modes vanishes at the MCT crossover temperature. At any temperature, however, saddles also possess a finite fraction of localized unstable modes, which only involve a finite number of particles, as originally envisaged by Goldstein. The extent to which the MCT transition is avoided in actual supercooled liquids thus depends on the concentration of such localized excitations [26].

Despite these advances, the spatial structure of the unstable modes remains largely unknown. In fact, the simple finite size scaling analysis of Ref. [26] was only based on the bulk average mode participation and further investigation is needed to characterize the modes' structure. Moreover, since the saddle modes progressively stabilize at low temperature, it is natural to inquire the connection between them and the stable modes populating the low-frequency portion of the vibrational spectrum. The lowest-frequency modes of local minima display quasi-localized vibrations (QLVs) [29,30], which are at present much better understood than the unstable saddle modes [29–37]. QLVs consist of an energetically unstable core and a stable far-field [38] and their eigenenergy is determined by the competition between these two components. The vibrational amplitudes decay with a distinct power law $r^{-(d-1)}$, where $r$ is the distance from the core and $d$ is the spatial dimension, unless hybridization with acoustic modes occurs [29,33,39]. Moreover, the QLVs spatial structure brings useful insight into several local phenomena in low temperature glasses, such as the response to a local dipolar force or plastic events associated to shear-transformation zones [40–45].

In this work, we apply the spatially resolved mode analysis developed to study QLVs [29, 38] to account for the saddle modes spatial structure. To achieve this, we improved the optimization protocol used in Ref. [26], which suffered from poor convergence when searching for stationary points in moderately large system sizes. Through this analysis, we confirm that the delocalized unstable modes that populate the saddles' spectrum at high temperature have an extended character, at least up to the length scale we could probe. We also show that the localized unstable modes are unique to the saddle structure. Their cores consist of only a few particles displaying very large displacements, while their far field decays rapidly compared to that of delocalized modes and of the QLVs. In local minima, truly localized vibrations would only be found in the high-frequency tail of the spectrum. However, we provide evidence that

around and below $T_{\mathrm{MCT}}$ localized unstable modes originate from structural defects similar to those associated to the cores of the QLVs, or "soft spots", which have been identified in previous studies on metallic glasses [46]. Overall, our study casts a bridge between so far disconnected investigations of the PES and provides a spatially resolved picture of the progressive stabilization of saddle modes as a supercooled liquid turns into a glass.

This paper is organized as follows. In Sec. 2.1, we introduce our model and describe how we located saddles and local minima starting from equilibrium configurations. In Sec. 2.2 and 2.3.1, we define the quantities used to characterize these stationary points. We provide our main results in Sec. 3. We focus on the spatial structure of unstable saddle modes in Sec. 3.1 and the local structure of unstable cores in Sec. 3.3. We compare saddles and local minima in Sec. 3.2. Finally, we conclude our work with a summary in Sec. 4.

# 2 Methods

## 2.1 Sample preparation

We present numerical results for the ternary mixture model introduced by Gutiérrez *et al.* [47]. The model consists of three species of particles, which interact via a repulsive inverse power potential with exponent 12

$$u_{\alpha\beta}(r) = \epsilon \left( \frac{\sigma_{\alpha\beta}}{r} \right)^{12} + c_4 \left( \frac{\sigma_{\alpha\beta}}{r} \right)^{-4} + c_2 \left( \frac{\sigma_{\alpha\beta}}{r} \right)^{-2} + c_0 \,, \tag{1}$$

where $\alpha, \beta = A, B, C$ are species indices. The additional terms in Eq. (1) ensure continuity of the potential up to its second derivative at the cutoff distance $r_{cut} = 1.25\sigma_{\alpha\beta}$. See Ref. [48] for the expressions of $c_0$, $c_2$, and $c_4$. The interactions are additive, the size ratio is $\frac{\sigma_{AA}}{\sigma_{BB}} = \frac{\sigma_{BB}}{\sigma_{CC}} = 1.25$ and the chemical compositions are $x_A = 0.55$, $x_B = 0.30$, and $x_C = 0.15$. In the following, we will express energies and distances in units of $\epsilon$ and $\sigma_{AA}$, respectively.

The equilibrium configurations we used for this study were obtained using swap Monte Carlo simulations in Ref. [26]. In the following, we will focus on systems composed of $N = 3000$ particles at a number density $\rho = 1.1$ and temperatures ranging from 0.45 to 0.28. Note that a few samples crystallized at $T = 0.28$ and were therefore removed from the analysis. Results for $N = 1000$ are shown in Appendix C. The mode-coupling critical temperature, estimated in Ref. [49] by fitting the structural relaxation times to a power law, is $T_{\mathrm{MCT}} \approx 0.29$. Starting from these equilibrium configurations, we performed three kinds of optimizations: (i) potential energy minimizations using the l-BFGS method [50] to locate the local minima of the PES; (ii) total square-force minimizations (or $W$-minimizations) using the l-BFGS method to locate local minima of

$$W = \frac{1}{N} \sum_{i=1}^{N} |\vec{F}_i|^2 \,, \tag{2}$$

where $\vec{F}_i$ is the force acting on particle $i$; (iii) eigenvector-following (EF) optimizations [51] to locate stationary points of the PES with a prescribed number $n_u$ of unstable modes.

Our goal is to locate stationary points, either local minima or saddles, in the neighborhood of a given equilibrium configuration. In both cases, stationary points are identified as points for which $W$ is smaller than $10^{-10}$. As is well known, $W$-minimizations mostly locate so-called quasi-saddles, i.e., local minima of $W$ with a finite value of the total force and precisely one inflection mode, while only a small fraction of $W$-minimizations reach true stationary points. By contrast, the EF method is guaranteed to converge to a stationary point of prescribed order $n_u$, but the value of $n_u$ has to be provided as input. To define a mapping between a given

configuration and its neighboring stationary point we have followed the approach used in Ref. [26]: for each starting configuration, we first perform a $W$-minimization. The order $n_u$ of the $W$-minimized configuration is then used as the target order for an EF optimization. All the saddles analyzed in Sec. 3 were obtained using this protocol and converged to the required tolerance, $W < 10^{-10}$.

The EF optimizations carried out in Ref. [26] displayed poor convergence for the system size of interest in this work ($N = 3000$). One possible reason is that the algorithm gets stuck in some restricted portion of configuration space, without being able to reach the neighborhood of the stationary point, where convergence is fast. To help the algorithm escape faster from those regions, in this work we have allowed for larger steps in configuration space. In the EF optimization method, the size of the steps is limited by a set of trust radii [51], which ensure a local harmonic approximation at each step. To mitigate the convergence issues, we have therefore used a larger tolerance on the deviation from harmonic approximation, namely we increased (decreased) the trust radii by 20% if the relative error on the corresponding eigenvalue was smaller (larger) than 200%. The threshold was 100% in Ref. [26]. Also, we set the initial trust radius to 0.1, instead of 0.2. Empirically, we found that using a larger tolerance substantially improved the convergence rate for large system sizes. The fraction of optimizations that converged to a stationary point after 4000 iterations ranged from 100% at $T = 0.28$ to 56% at $T = 0.45$. The fractions obtained using a threshold of 100% ranged from 48% at $T = 0.28$ to 3% at $T = 0.35$. Of course, using a large tolerance sometimes led to substantial overlaps between particles and thus large values of the potential energy. To prevent numerical issues, we avoided steps such that the potential energy of the new configuration was larger than a threshold ($10^4$), and decreased all the trust radii until the potential energy of the new configuration dropped below the threshold. Except for these small but important practical details, the algorithm was the same as the one used in Ref. [26].

## 2.2 Normal mode analysis

We performed a standard normal mode analysis [52] for local minima and saddles. We diagonalized the dynamical matrix, which is the Hessian of the total potential energy $U$,

$$\mathcal{H}_{ij} = \frac{\partial^2 U}{\partial \vec{r}_i \partial \vec{r}_j} = \begin{cases} -\mathcal{M}_{ij} & (i \neq j) \\ \sum_{k=1}^{N} \mathcal{M}_{ik} & (i = j) \end{cases}, \tag{3}$$

where

$$\mathcal{M}_{ij} = u_{ij}''(r_{ij}) \vec{n}_{ij} \vec{n}_{ij}^T + \frac{u_{ij}'(r_{ij})}{r_{ij}} \left( I - \vec{n}_{ij} \vec{n}_{ij}^T \right), \tag{4}$$

is the contribution from the pair $\langle ij \rangle$, $u_{ij}(r_{ij})$ is the pair interaction potential, $\vec{n}_{ij} = \vec{r}_{ij}/r_{ij}$ is the unit vector along the pair, and $I$ is the $d \times d$ identity matrix. We denote its eigenvalues by $\lambda_\alpha$ and the corresponding eigenvectors by $\vec{e}_\alpha = (\vec{e}_{\alpha,1} \cdots \vec{e}_{\alpha,N})$, where $\alpha = 1, 2, \ldots, 3N - 3$. We sort them in ascending order as $\lambda_1 < \lambda_2 < \ldots$. Note that we always excluded the three eigenmodes corresponding to global translations. Using these eigenmodes we computed the quantities defined below.

### 2.2.1 Participation ratio

The participation ratio [53–55] of a mode $\alpha$ is given by

$$P(\lambda_\alpha) = \frac{1}{\sum_{i=1}^{N} |\vec{e}_{\alpha,i}|^4}. \tag{5}$$

It quantifies the degree of localization of a given mode. When all particles have equal displacements, $P(\lambda_\alpha) = N$, while $P(\lambda_\alpha) = 1$ when the mode is localized on a single particle.

### 2.2.2 Decay profile

The decay profile [29, 39] $d_\alpha(r)$ of a mode $\alpha$ is a function of the distance $r$ from the particle with the largest $|\vec{e}_{\alpha,i}|$, which we denote by $i_d$. The decay profile is defined as

$$d_\alpha(r) = \underset{r_i \in [r - \Delta r/2, r + \Delta r/2]}{\text{median}} |\vec{e}_{\alpha,i}|, \tag{6}$$

where $r_i$ is the distance of a particle $i$ from the particle $i_d$. In the case of saddles, we show the averaged decay profile for a certain eigenvalue $\lambda$ defined as

$$d(r) = \langle d_\alpha(r) \rangle_\lambda, \tag{7}$$

where $\langle \bullet \rangle_\lambda$ denotes the average over all the eigenmodes with eigenvalues $\lambda_\alpha \in [\lambda - \Delta\lambda/2, \lambda + \Delta\lambda/2]$. For local minima, we averaged the data over the lowest frequency mode of each sample, which are almost always QLVs for $N \lesssim 10^4$ [29].

### 2.2.3 Fractal dimension

To further characterize the spatial structure of the eigenmodes, we introduce another function of the norms $|\vec{e}_{\alpha,i}|$. We identify the particles contributing to a mode $\alpha$ as those with the $\lceil P(\lambda_\alpha) \rceil$ largest norms, where $\lceil x \rceil$ denotes the ceiling function, which returns the least integer greater than or equal to $x$, and then determine the number $N_\alpha(r)$ of such contributing particles up to a distance $r$ from the particle $i_d$. We define the averaged function $N(r)$ by the same procedure as in Eq. (7). From this function, we can estimate the fractal dimension $D$ of mode of eigenvalue $\lambda$ from $N(r) \sim r^D$.

### 2.2.4 Energy profile

We define the local vibrational energy of a particle $i$ on mode $\alpha$ as

$$\begin{aligned}
\delta E_{\alpha,i} &= \frac{1}{2} \sum_{j=1}^{N} \left( \vec{e}_{\alpha,i} - \vec{e}_{\alpha,j} \right)^T \mathcal{M}_{ij} \left( \vec{e}_{\alpha,i} - \vec{e}_{\alpha,j} \right). \\
&= \frac{1}{2} \sum_{j=1}^{N} \left[ u''_{ij}(r_{ij})(\vec{n}_{ij} \cdot \vec{e}_{\alpha,ij})^2 + \frac{u'_{ij}(r_{ij})}{r_{ij}} (\vec{e}^{\perp}_{\alpha,ij})^2 \right],
\end{aligned} \tag{8}$$

where $\vec{e}_{\alpha,ij} = \vec{e}_{\alpha,i} - \vec{e}_{\alpha,j}$ is the relative displacement between the pair and $(\vec{e}^{\perp}_{\alpha,ij})^2 = (\vec{e}_{\alpha,ij})^2 - (\vec{n}_{ij} \cdot \vec{e}_{\alpha,ij})^2$ is the squared transverse relative displacement. The energy profile [38] $\Lambda_\alpha(r)$ of a mode $\alpha$ is a function of the distance $r$ from the particle with the smallest $\delta E_{\alpha,i}$, which we denote by $i_e$. $\Lambda_\alpha(r)$ is defined as

$$\Lambda_\alpha(r) = \sum_{r_i < r} \delta E_{\alpha,i}, \tag{9}$$

where $r_i$ is the distance of a particle $i$ from the particle $i_e$. This measures the vibrational energy of the system when a given mode is excited, and we do not include the total potential energy of saddles or minima. In the following, we show the averaged normalized energy profile defined as

$$\tilde{\Lambda}(r) = \frac{\langle \Lambda_\alpha(r) \rangle_\lambda}{\left| \langle \lambda_\alpha \rangle_\lambda \right|}, \tag{10}$$

---

The distance $r$ used here differs slightly from the one used in the paper by Gartner and Lerner [39]. They defined the center of an eigenmode using $w$ particles with the largest $|\vec{e}_{\alpha,i}|^2$ and measured the distance $r$ from this center. They mainly used $w = 4$ while our definition corresponds to $w = 1$. This difference does not affect the our results qualitatively.

for saddles, while we used only the lowest frequency QLVs for local minima. Since $\sum_i \delta E_{\alpha,i} = \lambda_\alpha$, $\tilde{\Lambda}(r) \to -1$ as $r \to \infty$ when $\lambda < 0$ while $\tilde{\Lambda}(r) \to 1$ when $\lambda > 0$. Finally, we note that the difference between $i_d$ and $i_e$ does not matter in practice, because there is a negative correlation between $|\vec{e}_{\alpha,i}|$ and $\delta E_{\alpha,i}$ (see Appendix A).

### 2.2.5 One-particle dynamical matrix

Finally, using $\mathcal{M}_{ij}$ in Eq. (4), we define the one-particle dynamical matrix as

$$\mathcal{H}_{ij}^{(1)} = \delta_{ij} \sum_{k=1}^{N} \mathcal{M}_{ik}. \tag{11}$$

Clearly, the full dynamical matrix reduces to $\mathcal{H}_{ij}^{(1)}$ when only a tagged particle is allowed to move and the others are kept fixed. We denote its eigenvalues by $\mu_\beta$ and the eigenvectors by $\vec{f}_\beta$, where $\beta = 1, 2, \ldots, Nd$.

## 2.3 Local structure

### 2.3.1 Locally favored structures

To provide further insight into the localization features of the modes and their link to the local structure, we performed a radical Voronoi tessellation of the minimized configurations using the Voro++ software [56]. The radical Voronoi tessellation requires as input the typical diameters $r_\alpha$ of the particles of type $\alpha$, which we estimated from the positions of the first peaks of the partial radial distribution functions $g_{\alpha\alpha}(r)$ measured for minimized configurations at $T = 0.28$, namely $r_1 = 1.34$, $r_2 = 1.11$, $r_3 = 0.93$. Following a well-established approach [57], we characterized the shape the Voronoi polyhedra using the signature $(n_3, n_4, n_5, \ldots)$, where $n_k$ is the number of faces of the polyhedron having $k$ vertices. We found that the (0,0,12) signature, corresponding to an icosahedral local arrangement, becomes the most frequent at low temperature, in both minimized and instantaneous configurations. Following Ref. [24], we therefore identify the icosahedron as the locally favored structure of the model.

### 2.3.2 Structural order parameter

To provide further insight into the local structure of our ternary mixture model, we calculated the Tong-Tanaka structural order parameter $\Theta$ [58]. This order parameter measures the average local deviation from close packing of neighboring particles, and has been shown to correlate quite strongly with the local dynamics in some models of supercooled liquids [58].

For the calculation of $\Theta$ we followed the method described in Ref. [58], except for the following minor modification. Since the contact radius between two neighboring particles is defined in terms of the nominal radii $(\sigma_{ii} + \sigma_{jj})/2$, in our soft sphere model negative deviations from ideal packing can occur. Negative deviations are harmless, as they only imply a linear shift of the order parameter, provided the absolute value in Eq. 2 of Ref. [58] is removed. We checked that this minor modification, which we used in this work, does not affect our calculation. In particular, at $T = 0.28$, reducing the contact radii by about 10% with respect to their nominal values effectively removes all negative contributions to $\Theta$ and lead to qualitatively similar results. Finally, we note that as in Ref. [58], we identified the network of neighbors using a radical Voronoi tessellation, see Sec. 2.3.1.

## 3 Results

### 3.1 Spatial structure of unstable modes

Table 1: Range of eigenvalues used to compute the average in Eq. (7), mobility edge $\lambda_e$, and fraction of the delocalized unstable modes $n_d/(3N)$ at all investigated temperatures. Note that a few isolated eigenvalues are included in the averages at $T = 0.28, 0.39, 0.32$.

| $T$ | $-\lambda$ | $-\lambda_e$ | $n_d/3N$ |
|------|-----------------------|--------------|-----------|
| 0.45 | 1.25–23.75 | 10.9 | 0.015 |
| 0.35 | 1.25–21.25 | 4.92 | 0.005 |
| 0.32 | 1.25–21.25, 36.25 | 2.71 | 0.0018 |
| 0.30 | 1.25–18.75, 26.25 | 1.40 | 0.00048 |
| 0.28 | 1.25–26.25, 31.25 | – | – |

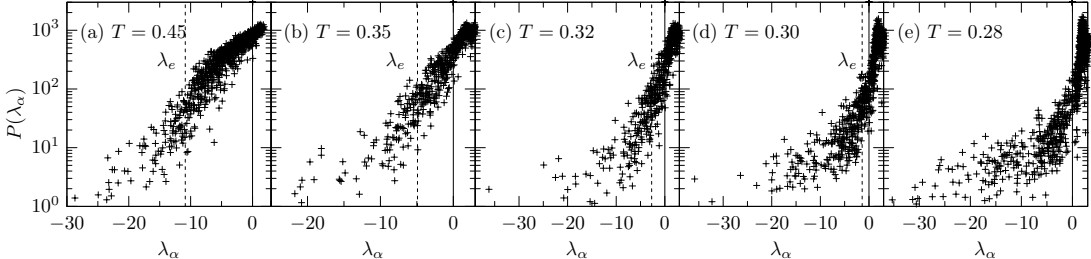

Figure 1: Scatter plots of the participation ratio for saddles at (a) $T = 0.45$, (b) $T = 0.35$, (c) $T = 0.32$, (d) $T = 0.30$, and (e) $T = 0.28$. In each panel, the solid and dotted lines correspond to $\lambda = 0$ and to the mobility edge $\lambda_e$, respectively.

In this section, we quantitatively characterize the spatial structure of the unstable saddle modes across the MCT crossover temperature. We show the corresponding results for the QLVs of the local minima in Appendix B, but some of the data of the QLVs are included in this section for comparison.

To briefly summarize the results of Ref. [26] using our data, we show the scatter plots of the participation ratio in Fig. 1. We show the results at (a) $T = 0.45$, (b) $T = 0.35$, (c) $T = 0.32$, (d) $T = 0.30$, and (e) $T = 0.28$. From a finite-size scaling analysis of the participation ratio, we can define the mobility edge $\lambda_e$ that separates localized and delocalized modes. It is the fixed point in $P(\lambda, N)N^{-1/3}$, where $P(\lambda, N)$ is the average participation ratio of modes with eigenvalue $\lambda$ in a system of size $N$. The modes below the mobility edge are localized because their participation ratio is generally independent of $N$, whereas the modes above the mobility edge are delocalized because their participation grows, on average, at least linearly with the linear system size $N^{1/3}$. This definition of the mobility edge was originally given in the context of the Anderson localization [59] and later adapted to the study of instantaneous normal modes by Clapa *et al.* [60]. The mobility edges determined in Ref. [26] using the above procedure are shown as dotted lines in Fig. 1; note that $\lambda_e$ vanishes around $T_{MCT} \approx 0.29$. Table 4 shows the specific values of $\lambda$ and $\lambda_e$ we used to perform the average in Eq. (7). We used a bin width $\Delta\lambda = 2.5$ and ignored the bins containing only one mode.

Before proceeding to a quantitative analysis of our data, we visualize the real space structure of some selected unstable modes, to grasp the main difference between localized and

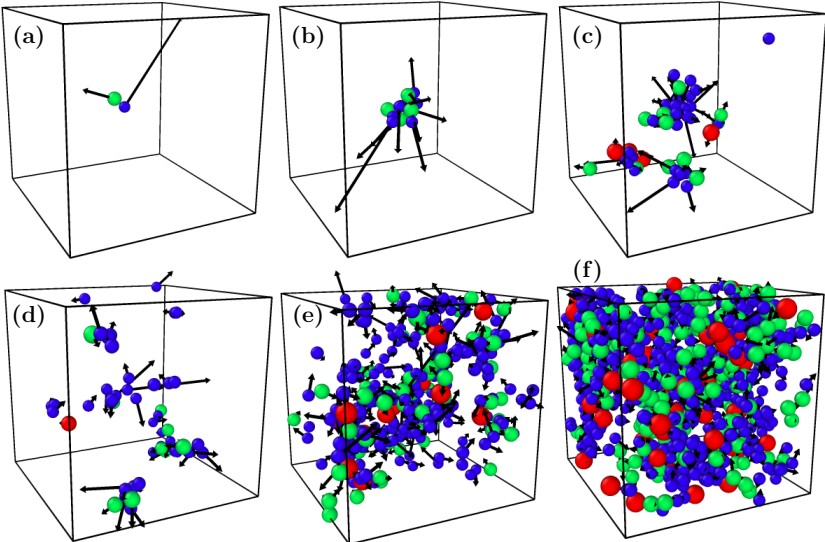

Figure 2: The unstable modes of a saddle at $T = 0.45$. We show $\lceil P(\lambda_\alpha) \rceil$ particles with the largest norms shown by black arrows. The particle with the largest norm is placed at the center in each box. Different colors indicate different particle types. We scale the norms by factors 20 in (a), (b), (c), and (d), and 40 in (e) and (f) for visualization purposes. The parameters of each mode are as follows: (a) $\alpha = 1$, $\lambda_\alpha = -28.7$, and $\lceil P(\lambda_\alpha) \rceil = 2$, (b) $\alpha = 11$, $\lambda_\alpha = -13.3$, and $\lceil P(\lambda_\alpha) \rceil = 15$, (c) $\alpha = 21$, $\lambda_\alpha = -10.6$, and $\lceil P(\lambda_\alpha) \rceil = 45$, (d) $\alpha = 41$, $\lambda_\alpha = -7.72$, and $\lceil P(\lambda_\alpha) \rceil = 49$, (e) $\alpha = 81$, $\lambda_\alpha = -4.79$, and $\lceil P(\lambda_\alpha) \rceil = 275$, and (f) $\alpha = 151$, $\lambda_\alpha = -0.722$, and $\lceil P(\lambda_\alpha) \rceil = 815$.

delocalized modes. In Fig. 2 we show six modes of a typical saddle configuration at $T = 0.45$. In each box, we only show the $\lceil P(\lambda_\alpha) \rceil$ particles having the largest norms. Different colors indicate different particle types. The particle with the largest norm is placed at the center of the box and the displacements are shown by black arrows. We scale the norms by a factor 20 in (a), (b), (c), and (d), and 40 in (e) and (f) for visualization purposes. Details of the parameters used for each mode are given in the caption of Fig. 2. Here, we note that the modes shown in (a) and (b) are localized and that the others are delocalized. We can clearly see the cores of the localized modes in Fig. 2(a) and (b), while it is difficult to identify similar cores in the delocalized modes, at least by visual inspection. The snapshots of Fig. 2 are also suggestive of the presence of a percolation transition as $P$ increases. In a first attempt to address this question, we have analyzed the network of the $[P]$ particles which, for a given mode, have the largest participation ratios. Our preliminary results are qualitatively compatible with a continuum percolation at a threshold participation ratio in the range $300 - 400$, but the present system size is obviously too small to draw firm conclusions. We leave this interesting point to a future investigation.

Let us start our analysis by looking at the decay profiles. We remind that the decay profiles of QLVs in local minima display a $r^{-2}$ scaling away from the cores [29], which is the same as the response of the elastic body to a local force dipole. We group the modes according to their eigenvalues as described in Sec. 2.2.2 and analyze the same set of temperatures as in Fig. 1. The results are shown in Fig. 3. The continuous change in color from blue to red corresponds to the change of $\lambda$ shown in Tab. 4: the data above the mobility edge in red and those below the mobility edge in blue. For comparison, we also plot the data for the QLVs in black. In our data for the QLVs, we cannot observe a clear asymptotic behavior $d(r) \propto r^{-2}$ because our

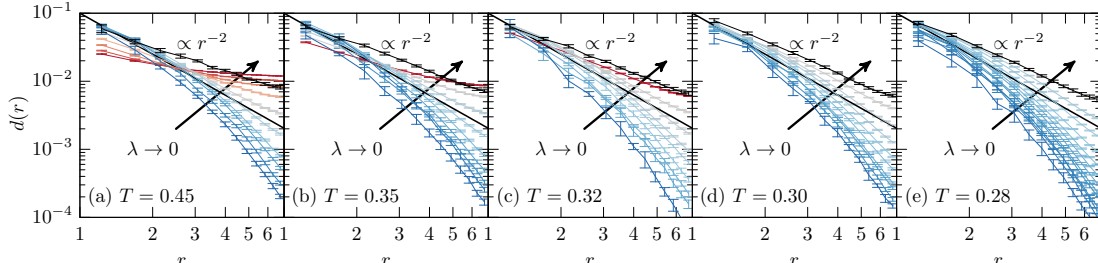

Figure 3: Decay profiles for saddles at (a) $T = 0.45$, (b) $T = 0.35$, (c) $T = 0.32$, (d) $T = 0.30$, and (e) $T = 0.28$. The solid lines are proportional to $r^{-2}$. The continuous change in color from blue to red corresponds to the change of $\lambda$ shown in Tab. 4. We show the data above the mobility edge in red, those below the mobility edge in blue. For comparison, we show the data the QLVs in black.

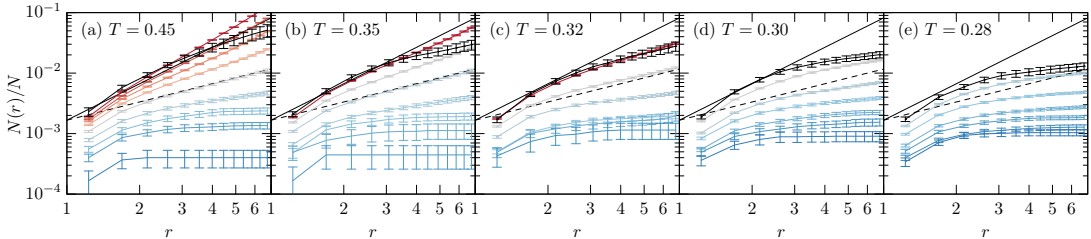

Figure 4: The fraction of particles $N(r)/N$ up to a distance $r$ for unstable modes of saddles. All parameters are the same as in Fig. 3, but we do not show the data with $\lambda < -21.25$ at $T = 0.45$, $\lambda < -16.25$ at $T = 0.35$, and $\lambda < -13.75$ at $T \leq 0.32$ for visualization purposes. The dashed and solid lines show power laws with the fractal dimensions $D = 1$ and 2, respectively.

systems are too small for this purpose [†]. Nonetheless, all the panels show the same tendency: $d(r)$ decreases more rapidly than $r^{-2}$ at large $r$ for $\lambda \ll \lambda_e$, while $d(r)$ tends to saturate at finite values at large $r$ for $\lambda \gg \lambda_e$. The former implies that the unstable modes much below the mobility edge are definitely more localized than the QLVs of minima, while the latter that the unstable modes much above the mobility edge are definitely delocalized, which cannot be explained as the response of the elastic body to a local disturbance. Close to the mobility edge ($\lambda \sim \lambda_e$), unstable modes display $d(r) \sim r^{-2}$ in a wide region of $r$. However, the current system size is too small to clearly identify this scaling as the asymptotic behavior. To achieve this, we would need to study much bigger systems, which are beyond the reach of this work.

The delocalized modes can be better characterized by $N(r)/N$, which allows us to identify the average fractal exponent of the modes. We show these results in Fig. 4. All parameters are the same as in Fig. 3, but for visualization purposes we do not show the data with $\lambda < -21.25$ at $T = 0.45$, $\lambda < -16.25$ at $T = 0.35$, and $\lambda < -13.75$ at $T \leq 0.32$. The dashed and solid lines show power laws with the fractal dimensions $D = 1$ and 2, respectively. This figure shows that the modes with $\lambda \sim \lambda_e$ (shown in gray) follow $N(r) \sim r^1$. This is consistent with the definition of the mobility edge, at which $P$ grows linearly with $L$. We note, however, that $N(r)$ is an averaged quantity and there are strong mode-to-mode fluctuations (not shown). Thus, we should not expect that each mode at the mobility edge has a string-like structure, as one may naively guess. This is also clear from Fig. 2(c), which shows a mode at the mobility edge having a complex, though somewhat ramified, spatial structure. As for the other modes, the curves of the localized modes rapidly converge to their final values while the delocalized modes and the

---

[†]In Ref. [29], systems of $10^6$ particles were needed to establish this scaling.

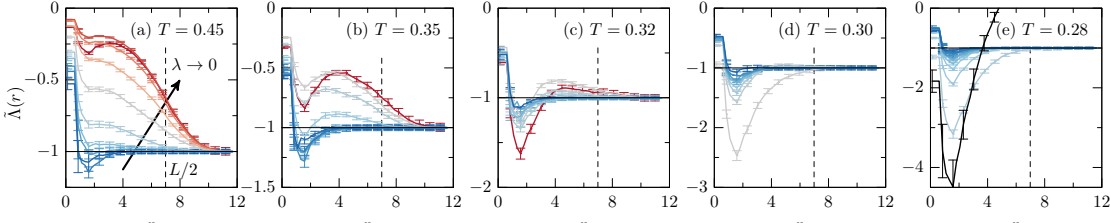

Figure 5: Energy profiles for saddles. All parameters are the same as in Fig. 3. We show the data of the QLVs only in (e). The dashed vertical lines show half of the box length $L/2$.

QLVs grow faster than linear with $r$. Nevertheless, we can also see a small difference between the delocalized unstable modes and the QLVs, particularly at $T \geq 0.35$: the latter saturate at large distances. Although subtle, this difference shows that the particles contributing to the QLVs are denser and more compact than those contributing to the delocalized modes. This observation is consistent with the results of the decay profiles.

To further characterize the mode structure, we analyze the energy profiles, see Fig. 5. The parameters are the same as in Fig. 3, but we show the data of the QLVs only at $T = 0.28$ because they display very little temperature dependence (see Appendix B). The dashed vertical lines are placed at half of the box length $L/2$. We find that localized and delocalized unstable modes display qualitatively different energy profiles. The former have pronounced dips at $r \sim 2$, which are the cores of the modes [32], and for $r \gtrsim 2$ they rapidly converge to $-1$ from below. Thus, the localized modes have weak far-field components and their energy is mostly determined by the cores. This is consistent with the results of the decay profile and differs markedly from the QLVs, whose energy is determined by the competition between the unstable core and the stable far-field. We also note that the energy profiles of the localized modes do not depend on the temperature above the MCT crossover temperature. By contrast, the delocalized modes have the large far-field components and display a pronounced temperature dependence. The energy profiles of the delocalized modes also show a small dip (or kink) at $r \sim 2$, which indicates that even these modes may possess a core. However, at $T \geq 0.32$ we observe $\tilde{\Lambda}(r \sim 2) > -1$. Therefore, the energy profiles converge to $-1$ from above. In other words, both the core and the far field are unstable. This is significantly different from the localized modes and also from the QLVs.

For $T \leq 0.30$, i.e. below the localization transition, the energy profiles converge to $-1$ from below and the functional form is similar to that of the QLVs. This indicates that the softest localized unstable modes that survive below the mode-coupling crossover [26] have spatial properties similar to the soft stable modes, see also Sec. 3.3. We emphasize, however, that the energy profiles discussed here characterize only the *average* spatial features of the vibrational modes. Dips or kinks at $r \sim 2$ in the energy profiles indicate the presence of regions that are particularly unstable on average. Our analysis does not tell, however, how many cores a given mode possesses. Actually, since delocalized modes have unstable far fields at high temperatures, one may speculate that they even have multiple cores. Only when the mode is strongly localized, as observed in Fig. 2(a) and in Fig. 6(a), the particle $i_e$ corresponds to the unique core of the mode. Finally, we expect that gathering the cores of the QLVs should provide similar information as the "soft spots" [40], although the two definitions differ in practice.

To better understand the results of the energy profiles, we visualize in Fig. 6 the spatial distribution of the local energy $\delta E_{\alpha,i}$ of the unstable modes. We used the same modes as in Fig. 2 and the particles with the most negative $\delta E_{\alpha,i}$ are placed at the center of the squares.

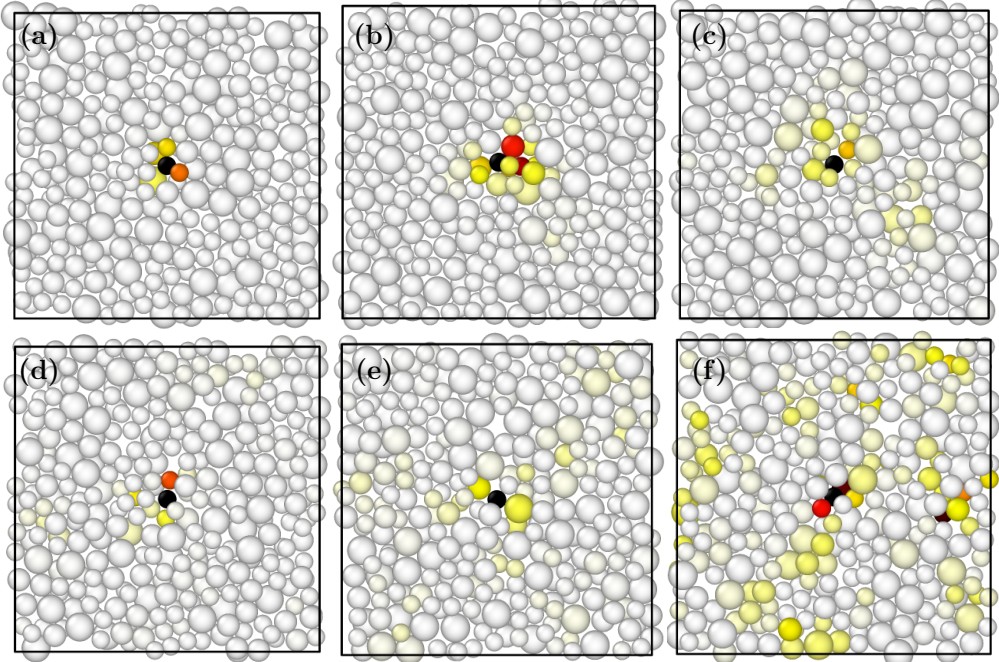

Figure 6: Spatial distribution of the local energy $\delta E_{\alpha,i}$ for the same modes as in Fig. 2. Each snapshot represents a slab of width 1.5. The particles with the most negative $\delta E_{\alpha,i}$ are placed at the center of the squares. The particles with darker colors have more negative energy, while particles with positive energy are colored white.

Particles having more negative energy are colored with darker colors, while those with positive energy are colored white, regardless of the value of $\delta E_{\alpha,i}$. We can see a few dark particles at the centers of the boxes in all cases; they correspond to the small dips at $r \sim 2$ observed in Fig. 5 (a). The localized modes with small $P(\lambda_\alpha)$ have energetically stable backgrounds, *i.e.*, the background is uniform white, while the delocalized ones with large $P(\lambda_\alpha)$ have unstable backgrounds, *i.e.*, there are scattered yellow regions. This is consistent with the differences between localized and delocalized modes mentioned above.

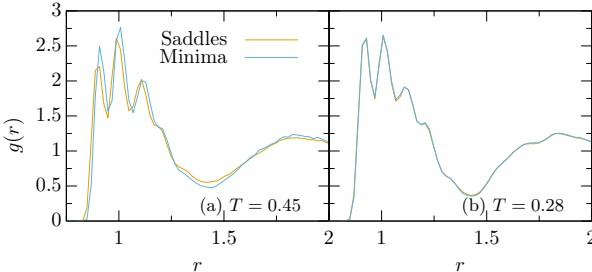

Figure 7: Radial distribution function for saddles and local minima of $N = 3000$. The temperatures are (a) $T = 0.45$ and (b) $T = 0.28$.

## 3.2   Local structure of saddles and local minima

In the previous section, we showed that the localized modes of saddles are generally more localized than the QLVs. Since the interaction potential is the same, it is plausible to expect that this reflects a structural difference between saddles and local minima. Otherwise, it may

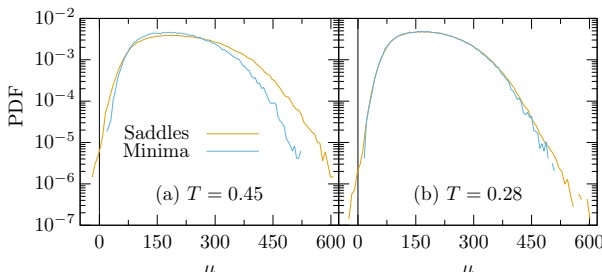

Figure 8: PDF of the eigenvalues of the one-particle dynamical matrix. All parameters are the same as in Fig. 7.

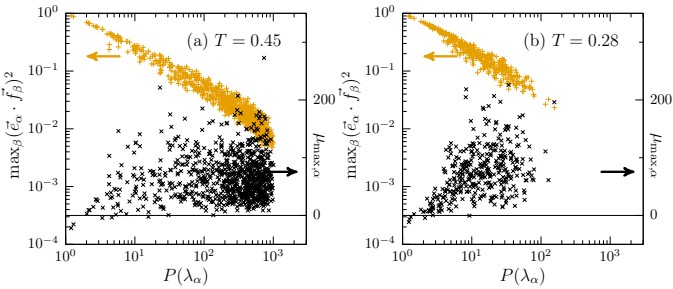

Figure 9: Left axis: the largest overlap between the eigenvectors of the full dynamical matrix and the one-particle dynamical matrix $\max_\beta(\vec{e}_\alpha, \vec{f}_\beta)^2$. Right axis: the corresponding eigenvalue $\mu_{\mathrm{max},\alpha}$. The abscissa is the participation ratio. We show the data for saddles at (a) $T = 0.45$ and (b) $T = 0.28$.

also be that unstable and stable modes are intrinsically different. To investigate this point, we computed the radial distribution function $g(r)$ for for saddles and local minima, see Fig. 7. From inspection of these functions, we cannot see any difference between saddles and local minima at low temperature, $T = 0.28$. By contrast, the first peaks of local minima shift outwards compared to that of saddles at $T = 0.45$. Thus, some particles of saddles are closer to each other than particles of local minima. The very existence of unstable modes in the saddles can be partly attributed to these structural differences. This is because the closer two particles, the larger the force, and as is evident from the second term in Eq. (4), a large repulsive force significantly decreases the eigenenergy of the modes. We note that such destabilization plays a dominant role also for QLVs [38]. In addition, the dip at $r \sim 1.4$ for local minima shifts slightly downwards at $T = 0.45$.

To further address this point, we characterize the local structure of the local minima and saddles using a Voronoi tessellation, see Sec. 2.3.1. As explained in that section, we characterized the local arrangements around each particle using the signature of corresponding Voronoi polyhedron, as is routinely done to identify the locally favored structures of supercooled liquids [24]. We found that the Voronoi signatures in saddles and minima have different statistics at $T = 0.45$. Local minima having a sensibly larger fraction of icosahedral structures, which we identified as the locally favored structure of the model. By contrast, the statistics of signatures become very similar for minima and saddles sampled at low temperature ($T = 0.28$), see Tab. 2, and the fraction of the icosahedral structures is only marginally larger in minima. This confirms that overall minima are locally more structured than saddles at high $T$, but that these structural differences fade away close to $T_{\mathrm{MCT}}$. The overall structural similarity between saddles and minima at low temperature suggests that the localized unstable modes find their origin in subtle features of the fluid structure.

Table 2: Percentage of most frequent Voronoi signatures in saddles and local minima sampled at $T = 0.28$. The $\star$ symbol denotes the signature with the largest occurrence for core particles.

| $T = 0.28$ | | | |
|---|---|---|---|
| Saddles | | Local minima | |
| (0,0,12) | 17.9 | (0,0,12) | 18.9 |
| (0,2,8,2) | 7.2 | (0,2,8,2) | 7.4 |
| (0,1,10,2) | 6.1 | (0,1,10,2) | 6.3 |
| (0,3,6,4)$^\star$ | 3.8 | (0,2,8,1) | 3.8 |
| (0,2,8,1) | 3.8 | (0,3,6,4)$^\star$ | 3.8 |
| (0,2,8,4) | 3.0 | (0,2,8,4) | 3.0 |
| (0,1,10,4) | 2.3 | (0,1,10,4) | 2.5 |
| (0,3,6,3) | 2.1 | (0,1,10,3) | 2.1 |
| (0,2,8,5) | 2.0 | (0,3,6,3) | 2.1 |
| (0,1,10,3) | 2.0 | (0,2,8,5) | 2.1 |

Table 3: Percentage of most frequent Voronoi signatures in saddles and local minima sampled at $T = 0.45$.

| $T = 0.45$ | | | |
|---|---|---|---|
| Saddles | | Local minima | |
| (0,2,8,2) | 4.7 | (0,2,8,2) | 7.2 |
| (0,3,6,4) | 4.4 | (0,3,6,4) | 5.3 |
| (0,2,8,1) | 3.3 | (0,2,8,1) | 4.7 |
| (0,3,6,3) | 3.2 | (0,0,12) | 4.6 |
| (0,1,10,2) | 2.5 | (0,1,10,2) | 4.2 |
| (0,2,8,4) | 2.4 | (0,3,6,3) | 3.5 |
| (0,0,12) | 1.9 | (0,2,8,4) | 3.2 |
| (0,3,6,5) | 1.6 | (0,2,8,5) | 1.9 |
| (1,2,6,3,1) | 1.6 | (0,3,6,5) | 1.8 |
| (1,2,5,4) | 1.5 | (0,2,8,3) | 1.7 |

## 3.3 Local structure of unstable and stable cores

One may expect that the unstable modes are localized around some sort of structural defects in the supercooled liquid, in analogy to what found in local minima [40, 46]. To address this question, we characterize the structure of the unstable modes' cores in three different ways: by analyzing the one-particle dynamical matrix, by computing restricted few-body correlation functions, and by analyzing the Voronoi cells surrounding the core particles.

The one-particle dynamical matrix is the dynamical matrix obtained when only a tagged particle is allowed to move and the others are kept fixed (see Sec. 2.2.5 for details). Here we show that its eigenmodes give us an insight into the cores of the localized unstable modes of saddles. In Fig. 8 we show the probability distribution functions (PDF) of its eigenvalues. All the parameters are the same as in Fig. 7. We can see a clear difference between saddles

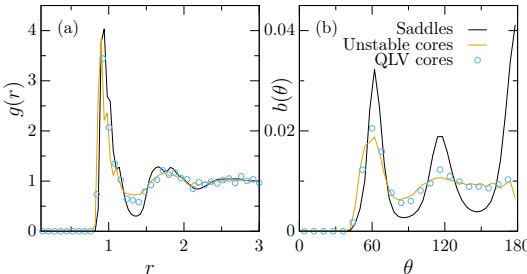

Figure 10: (a) Radial distribution function $g(r)$ and (b) bond-angle distribution functions $b(\theta)$ for small particles (black), for small particles at the core of unstable modes (orange) for saddles sampled at $T = 0.28$, and for small particles at the core of the QLVs (symbol) for local minima at the same temperature.

and local minima at $T = 0.45$, as the PDF of saddles has broader tails than that of local minima. Namely, some particles of saddles feel much steeper potentials in certain directions than particles of local minima. Thus, the structure of saddles is more inhomogeneous.

The localized unstable modes with small $P(\lambda_\alpha)$ have significant correlations with the eigenmodes of the one-particle dynamical matrix. Here, we consider the overlap between the eigenvectors of $\mathcal{H}$ and $\mathcal{H}^{(1)}$, i.e., $(\vec{e}_\alpha \cdot \vec{f}_\beta)^2$, to investigate whether the motions of the localized unstable modes can be explained at the one-particle level. Figure 9 shows the maximum overlap $\max_\beta (\vec{e}_\alpha \cdot \vec{f}_\beta)^2$ (left axis) and the corresponding eigenvalue $\mu_{\max,\alpha}$ (right axis) defined as

$$\mu_{\max,\alpha} = \mu_{\beta_{\max,\alpha}}, \tag{12}$$

where

$$\beta_{\max,\alpha} = \arg\max_\beta (\vec{e}_\alpha \cdot \vec{f}_\beta)^2. \tag{13}$$

The abscissa is the participation ratio $P(\lambda_\alpha)$ of the corresponding eigenmode of $\mathcal{H}$. We show the data of saddles at (a) $T = 0.45$ and (d) $T = 0.28$. This figure shows that the modes with smaller $P(\lambda_\alpha)$ have larger overlap with the eigenmodes of $\mathcal{H}^{(1)}$. Therefore, the motions of the strongly localized modes can actually be explained at the one-particle level. The correlation between $\mu_{\max,\alpha}$ and $P(\lambda_\alpha)$ is not strong, but $\mu_{\max,\alpha}$ tends to be negative when $P(\lambda_\alpha) < 2$.

To gain more insight into the structural features of the cores, we analyze selected few-body correlation functions. In this analysis, we focus on the local environment around particles of species C. This is motivated by the fact that the vast majority ($\approx 90\%$) of the core particles of the localized unstable modes belong to this species. Therefore, we restrict the calculation of the radial distribution function $g(r)$ to central particles of species C that form the cores of the unstable modes, *i.e.*, the particles whose index is $i_e$. Note that any other neighbors at a distance $r$ is used in this calculation, irrespective of its species. Similarly, we compute the restricted bond-angle distribution $b(\theta)$ obtained from the angles formed between a central particle of species C belonging to the cores and two of its nearest neighbors, irrespective or their species [61]. Note that the bond-angle distributions are normalized such that the distribution is flat when angles are drawn randomly on a sphere. We emphasize that these correlation functions do not entail information of the correlations *between* the cores but on the local structure around the core particles.

We perform our analysis at $T = 0.28$, where only localized unstable modes are present and the identification of the cores is therefore unambiguous. Figure 10 reveals that the local structure around the core particles is markedly different from the average. In particular, the $g(r)$ of the core particles displays a very flat first minimum, suggestive of a much less structured first coordination shell. The absence of the splitting of the second peak of $g(r)$ means that core

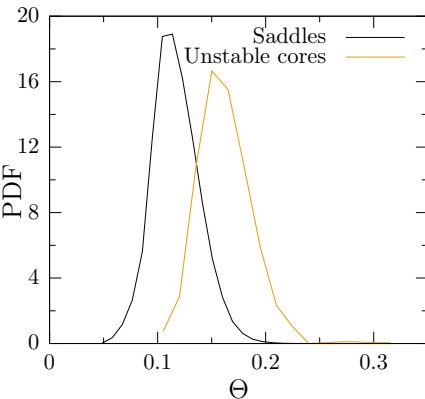

Figure 11: Probability density functions of the Θ order parameter for all particles in the saddles sampled at $T = 0.28$ (black line) and for particles belonging to the cores of the unstable modes (orange line).

particles are unlikely to align linearly with their neighbors [62], which is consistent with the absence of a peak at 180 degrees in the bond-angle distribution. Overall, the structure around core particles is almost featureless and qualitatively resembles the one of the fluid at a higher temperature. In particular, only at temperatures much higher than $T = 0.45$ the equilibrium $g(r)$ would display a first minimum as flat as the one found for core particles.

We also analyzed the signatures of the Voronoi cells surrounding the core particles and found that they never correspond to the locally favored structure of the model, *i.e.*, the icosahedron, which accounts for about 18% of the Voronoi signatures at $T = 0.28$, see Tables 2 and 3. The Voronoi cells around core particles are characterized instead by a broad distribution of signatures, not corresponding to any symmetric or clearly identified structural arrangement. The most frequent signature around core particles of unstable modes is (0,3,6,4) (4.6%), followed by (0,3,6,6) (2.6%), (0,4,4,4) (2.1%) and several others with similarly small concentrations. Interestingly, the cores of the QLVs in local minima display similar structural features: from Fig. 10 we see that their $g(r)$ and $b(\theta)$ resemble the ones of the unstable cores discussed above. Moreover, we found that the most frequent signature around core particles is the same for unstable modes and QLVs in local minima, albeit with a higher concentration (7.9%) in the latter. The remaining most frequent signatures around core particles in unstable modes and QLVs differ. Overall, the results indicate that, around and below the MCT crossover, localized unstable modes and soft stable modes have a similar structural origin.

To better identify the structural features of the unstable modes' cores, we calculated the Θ structural order parameter introduced in Ref [58], which measures the deviations from ideal close packings of spheres (see Sec. 2.3.2). We carried out this analysis at $T = 0.28$. We found that Θ is significantly larger for particles belonging to the unstable modes' cores than for the bulk, as demonstrated by the shift in the corresponding probability density $p(\theta)$, see Fig. 11. Therefore, the cores are associated to less compact local environments. We also calculated the precision with which Θ can identify the cores, as follows. Say we have $n_c$ core particles in a given saddle configuration. We then select the $n_c$ particles having the largest Θ values. The fraction of particles these subsets have in common defines the precision with which the cores are predicted by Θ. We found that the precision is higher than would be obtained by picking particles at random, but still relatively low ($< 20\%$). These preliminary results highlight the difficulty in pinpointing defects in glassy configurations. This problem may be partly attributed to the presence of spurious second shell neighbors in the radical Voronoi tessellation.

From this analysis we conclude that it is indeed appropriate to identify the core particles as "structural defects" in the liquid, in agreement with related observations about soft spots

in granular packings [40] and in a model metallic glass [46]. We note, however, that while the locally favored structure is the same in our model and in the one of Ref. [46], the Voronoi signatures of the structural defects have a very broad, model-specific distribution. Thus, this kind of Voronoi-based analysis does not clearly tell us what structural defects are – at best what they *are not*. Measurements of the Tong-Tanaka structural order parameter indicate that, in our model, unstable cores are localized around steric defects associated to loose local packings. However, it remains difficult to predict which particles will be associated to the cores. Devising a more general, unsupervised approach to identify structural defects is crucial to predict plastic events in glasses under shear [63] and also dynamic heterogeneities in supercooled liquids [58, 61, 64].

# 4 Conclusions

In this paper we have provided a spatially-resolved analysis of the structure of the unstable saddle modes of a ternary glass-former across the mode-coupling crossover. Our analysis provides an independent confirmation of the findings of Ref. [26] concerning the presence of two kinds of unstable modes: (i) delocalized modes, which are characterized by spatially extended displacement fields and which disappear below the mode-coupling crossover temperature $T_{\mathrm{MCT}}$ and (ii) localized modes, whose properties are largely independent of temperature. We have also performed a detailed comparison to the features of the softest stable modes observed in local minima. This extended analysis was possible thanks to some simple but crucial tweaks to the parameters of the EF optimization algorithm, which enabled us to study larger system sizes than those of Ref. [26].

Our results confirm that the disappearance of delocalized unstable modes below the MCT crossover is the only trace of the sharp change of the PES observed in mean-field glass models. By contrast, localized unstable modes find their origin in structural defects of the liquid, and are therefore a genuine finite-dimensional feature. The fact that these two kinds of modes coexist in a reasonably realistic model glass-former supports the view that the PES is a useful framework to reconcile mean-field and real space pictures of glasses. In particular, it would be worth investigating the connection between localized unstable modes and the localized excitations invoked in dynamic facilitated models and observed in some model glass-formers [65]. See also Ref. [66] for a recent attempt along similar lines using instantaneous normal modes.

These two kinds of modes are separated by the mobility edge $\lambda_e < 0$; the localized (delocalized) modes are found below (above) it, and $\lambda_e \to 0$ as $T \to T_{\mathrm{MCT}}$. Our independent analysis confirms the presence of a qualitative change of the spatial localization features across the mobility edge. We found that modes at the edge have an average fractal dimension of one, consistent with the finite size scaling analysis of Ref. [26]. However, we warn that one should not interpret this result in a naif geometric sense: unstable modes at the mobility edge have a very open structure but are not clearly string-like and the $N(r) \sim r$ scaling only holds *on average*. Our results call for a more in-depth characterization of the mode structure at a per-mode level.

We further compared the unstable modes of saddles to the QLVs that populate the soft portion of the vibrational spectrum of local minima. We found that at high temperature the local structure of local minima and saddles differ significantly, and the decay profiles and fractal dimensions of QLVs do not clearly match the ones of either delocalized or localized unstable modes. However, as the temperature drops around and below $T_{\mathrm{MCT}}$, local minima and saddles become structurally very similar and QLVs share some features with the softest unstable modes, *i.e.*, those with lowest absolute frequency. Our analysis of the energy profiles and of the structure of the modes' cores indicates that at low temperature unstable modes and

QLVs are localized around similar structural defects, but in the former the elastic response of the medium surrounding the core is not strong enough to stabilize the system. In the studied model, the unstable modes' cores are associated to steric defects, corresponding to loose local packings, localized outside the regions of local icosahedral order. Although the computational costs are far beyond our reach for the moment, it would also be useful to use much larger systems and investigate the precise functional forms of the decay and energy profiles of the saddle modes.

In conclusion, the PES of a simple but realistic model glass-former always possesses purely localized saddle modes, whose displacements and energies are concentrated on cores composed of few particles, i.e., they are defects unique to the saddle structure. By contrast, delocalized unstable modes, characterized by a slower decay of the displacement amplitudes and energy profiles, are only accessible at temperatures above the MCT crossover. At low temperature, soft unstable modes and QLVs share similar spatial features and they tend to be localized around similar structural defects. Our study connects previous investigations of the PES, which have focused either on high-order stationary points or on local minima, and provides a real space picture of how an equilibrium liquid characterized by several unstable modes converges to a mechanically stable glass.

## Acknowledgments

This work was supported by JSPS KAKENHI Grants No. 18H05225, 19J20036, 19H01812, 19K14670, 20H01868, 20H00128 and partially supported by Initiative on Promotion of Supercomputing for Young or Women Researchers, Supercomputing Division, Information Technology Center, The University of Tokyo. DC acknowledges support as a JSPS International Research Fellow.

## A Difference between $i_d$ and $i_e$

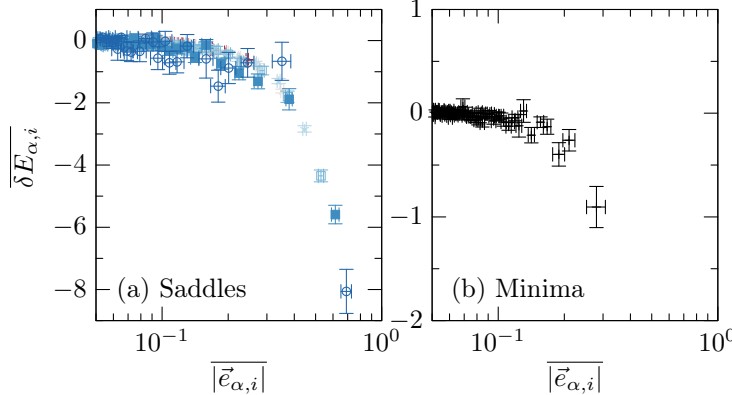

Figure 12: Scatter plots of $|\vec{e}_{\alpha,i}|$ vs. $\delta E_{\alpha,i}$ at $T = 0.32$ for (a) saddles and (b) local minima .

The origins of the functions $d(r)$ and $\Lambda(r)$ are different. However, it was shown in Ref. [38] using the QLVs of nearly jammed packings that there is a negative correlation between $|\vec{e}_{\alpha,i}|$ and $\delta E_{\alpha,i}$. We confirm this correlation in the present case. Figure 12 shows the data at $T = 0.32$. Different colors in (a) indicate different eigenvalues shown in Tab. 4. To average $|\vec{e}_{\alpha,i}|$ and

$\delta E_{\alpha,i}$, we sorted $|\vec{e}_{\alpha,i}|$ in descending order $|\vec{e}_{\alpha,1}| > |\vec{e}_{\alpha,2}| > \cdots > |\vec{e}_{\alpha,N}|$ and averaged $|\vec{e}_{\alpha,i}|$ and $\delta E_{\alpha,i}$ with fixed $i$.

## B   Local minima

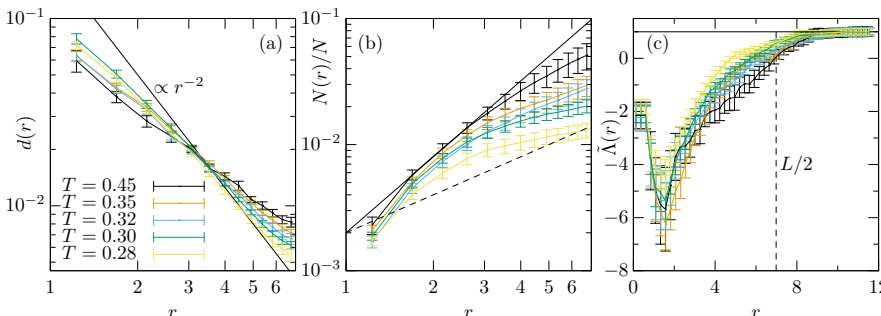

Figure 13: (a) Decay profile, (b) $N(r)/N$, and (c) energy profile for the QLVs of IS. The solid line in (a) is proportional to $r^{-2}$. The dashed and solid lines in (b) show the power laws with the fractal dimensions $D = 1$ and 2, respectively. The dashed vertical line in (c) shows half of the box length $L/2$.

To compare with the results of saddles, we show the results of local minima. The decay profiles, $N(r)/N$, and the energy profiles are shown in Fig. 13(a), (b), and (c), respectively. Different symbols and colors indicate different temperatures. Unlike the case of saddles, we averaged them over the lowest-frequency QLVs in each sample. We cannot see any strong temperature dependence, however the modes are slightly more localized when the temperature decreases.

## C   Size dependence

Table 4: Eigenvalues used to compute the average in Eq. (7), mobility edge $\lambda_e$, and fraction of the delocalized unstable modes $n_d/(3N)$ at all investigated temperatures for $N = 1000$ particles.

| $T$ | $-\lambda$ | $-\lambda_e$ | $n_d/3N$ |
|---|---|---|---|
| 0.45 | 1.25, 6.25, 11.25, 16.25, 21.25 | 10.9 | 0.015 |
| 0.35 | 1.25, 6.25, 11.25, 16.25, 21.25 | 4.92 | 0.005 |
| 0.32 | 1.25, 6.25, 11.25, 16.25 | 2.71 | 0.0018 |
| 0.30 | 1.25, 6.25, 11.25, 16.25 | 1.40 | 0.00048 |
| 0.28 | 1.25, 6.25, 11.25, 16.25 | – | – |

To study the system size dependence, we show the data for $N = 1000$. Figure 14 shows the corresponding decay profile. We show the data for $N = 1000$ by lines and those for $N = 1000$ by symbols. The specific eigenvalues used to compute the average are shown in Tab. 4. We can hardly see any size dependence. By contrast, the energy profiles do depend on system size, see Fig. 15. In particular, the delocalized unstable modes show a strong size dependence at high temperatures while the localized modes do not depend on $N$. This is natural because the

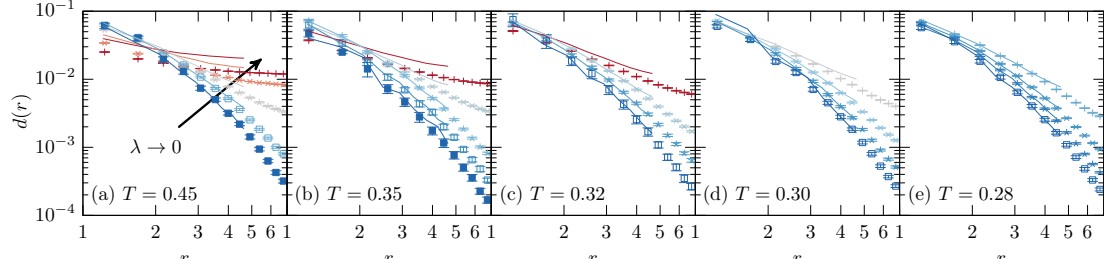

Figure 14: Finite size effects on the decay profile for saddles at (a) $T = 0.45$, (b) $T = 0.35$, (c) $T = 0.32$, (d) $T = 0.30$, and (e) $T = 0.28$. We show the data of $N = 1000$ by lines and those of $N = 3000$ by symbols. The continuous change in color from blue to red corresponds to the change of $\lambda$ shown in Tab. 4. We show the data above the mobility edge in red, those below the mobility edge in blue.

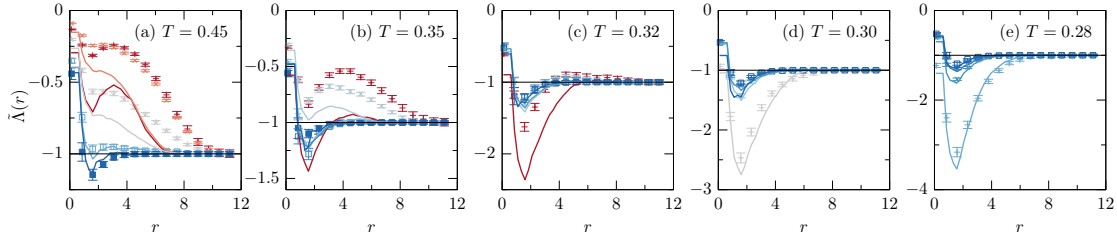

Figure 15: Finite size effects on the energy profile for saddles. All parameters are the same as in Fig. 14.

energy of the delocalized modes is determined by the entire system while the core dominates the energy of the localized modes. Note that, however, the qualitative behaviors do not depend on $N$ in any case.

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
