# Peer review of "Spatial structure of unstable normal modes in a glass-forming liquid"

_SciPost Physics, doi:SciPost Phys. 10, 001 (2021)_

## Round 1 · Referee Report · Anonymous (Referee 1) · 2020-10-16

Strengths

  • Clear
  • Insightful
  • Detailed

Weaknesses

No major weaknesses were identified.

Report

In their work “Spatial structure of unstable normal modes in a glass-forming liquid” the authors revisit the potential energy landscape picture of glassy liquids. Their study highlights the difference in the spatial distribution between unstable and stable modes revealing how the unstable modes change and become more localized as the temperature is decreased.

The article is very well written and clearly explained. I recommend publication but still invite the authors to consider the following remarks:

  1. In section IIA the authors state that a key difference between the present work and Ref26 is that using a larger tolerance improves the convergence, at the price of some occasional (unphysical?) configurations with exceedingly large energies. Could the authors expand on this point and attempt some (eventually heuristic) explanation of the meaning of the increased tolerance? What are the trade-offs? Why does it work?

  2. I am a bit confused by Fig 4: I suppose that the specific values of lambda are unimportant, and what matters is the asymptotic slope of the ration N(r)/N. As the temperature decreases, one is left only with modes below the mobility edge. It is not clear that these modes (the blue lines) are very sensitive to the temperature. The result seems to be that the localized saddles have very low fractal dimensions: can one infer from that they are “compact”? Isn’t it more appropriate to say that they are more point-like, as very few particles (or pairs of particles) for them?

  3. Along this line, it would be interesting if the authors could contrast their measure of populations of unstable cores with the excitations predicated in dynamical facilitation [Keyes PRX 2011]. Are the two notions incompatible?

  4. Regarding sections IIIB and C, it seems that the local environment (as probed by the radial distribution function or Voronoi tesselations) of saddles and local minima becomes more similar as the temperature is decreased. One then wonders how much of a ovarlap there is between the population of particles participating in the stable and unstable cores. Could one quantify this, for example with joint and conditional probabilities?

Requested changes

  1. page 3, last paragraph of section IIA: please clarify the definition of the rust radius
  2. Fig 1, both the two vertical lines are dashed (with different dash length) and not dashed and dotted as reported in the caption
  3. Fig 4, two values are needed to know the scale of for the x-axis. I suppose that the largest r value is around 7sigma?

  • validity: top
  • significance: high
  • originality: good
  • clarity: top
  • formatting: perfect
  • grammar: excellent

Author:  Masanari Shimada  on 2020-11-24  [id 1059]

(in reply to Report 1 on 2020-10-16)

In their work “Spatial structure of unstable normal modes in a glass-forming liquid” the authors revisit the potential energy landscape picture of glassy liquids. Their study highlights the difference in the spatial distribution between unstable and stable modes revealing how the unstable modes change and become more localized as the temperature is decreased.

The article is very well written and clearly explained. I recommend publication but still invite the authors to consider the following remarks:

We thank the referee for carefully reading and praising our manuscript. Below we addressed all the comments and questions, and we revised the manuscript accordingly.

In section IIA the authors state that a key difference between the present work and Ref26 is that using a larger tolerance improves the convergence, at the price of some occasional (unphysical?) configurations with exceedingly large energies. Could the authors expand on this point and attempt some (eventually heuristic) explanation of the meaning of the increased tolerance? What are the trade-offs? Why does it work?

We have slightly expanded the corresponding discussion in the Methods section. Essentially, poor convergence may occur if the algorithm gets stuck in some regions of configuration space, without being able to reach the neighborhood of the stationary point (where convergence is fast). If we tolerate larger steps (for which the local harmonic approximation is more strongly violated), the algorithm may escape more easily from those areas and find the stationary point. The drawback is that sometimes this produces larger values of the energy. When this occurs, we simply scale the step until the energy increase is below some threshold. One additional side-effect is that the algorithm may locate stationary points that are further away from the original configuration. Since we are only interested in the statistical properties of the stationary points, however, we do not think this is a serious issue.

I am a bit confused by Fig 4: I suppose that the specific values of lambda are unimportant, and what matters is the asymptotic slope of the ration N(r)/N. As the temperature decreases, one is left only with modes below the mobility edge. It is not clear that these modes (the blue lines) are very sensitive to the temperature. The result seems to be that the localized saddles have very low fractal dimensions: can one infer from that they are “compact”? Isn’t it more appropriate to say that they are more point-like, as very few particles (or pairs of particles) for them?

We checked the discussion of Fig.4 in our original submission and we confirm we did not describe the localized modes as "compact". We only described the QLV as "more compact" than the delocalized unstable modes. Actually, we agree with the referee that localized modes are nearly point-like and can therefore be mapped to "defects" in the supercooled liquid. We think this is already consistent with our current presentation.

Along this line, it would be interesting if the authors could contrast their measure of populations of unstable cores with the excitations predicated in dynamical facilitation [Keyes PRX 2011]. Are the two notions incompatible?

We agree with the referee that a comparison between unstable modes and dynamic excitations would be very interesting. Indeed, we pointed out this possibility already in the Conclusions of our initial submission. In this study, we decided to focus on the statistical properties of the PES within the harmonic approximation, for which many useful and well-defined methods are available. We thus prefer to leave a comparison with the dynamics to future work.

Regarding sections IIIB and C, it seems that the local environment (as probed by the radial distribution function or Voronoi tesselations) of saddles and local minima becomes more similar as the temperature is decreased. One then wonders how much of a ovarlap there is between the population of particles participating in the stable and unstable cores. Could one quantify this, for example with joint and conditional probabilities?

We thank the referee for this suggestion. We point out however that although searches for minima and saddles start from the same liquid configuration, the normal modes of the resulting configurations may be rather different from one another, and an overlap defined by an inner product between the modes, or conditional probabilities, will not be, in general, a good measure. The similarity observed in this study should be interpreted as an average property. A more sensible comparison of the modes may be done by searching for local minima connected to a given stationary point by a reaction path. This kind of analysis is left for a future study.

page 3, last paragraph of section IIA: please clarify the definition of the rust radius

We have added a brief clarification on the trust radii along with the appropriate reference (Ref. 52).

Fig 1, both the two vertical lines are dashed (with different dash length) and not dashed and dotted as reported in the caption

We have changed the line indicating lambda = 0 to a solid line.

Fig 4, two values are needed to know the scale of for the x-axis. I suppose that the largest r value is around 7sigma?

We thank the referee for this useful comment. We changed the format of x-axis.

---

## Round 1 · Referee Report · Anonymous (Referee 2) · 2020-10-18

Strengths

1) The manuscript is well written with clear figures and all technical details are well explained. 2) I find this article to provide interesting insights into the unstable modes in real space of finite dimensional liquids. 3) It is moreover shown that the localized unstable modes converge towards the quasi-localized soft modes commonly observed for minima as the temperature of the liquid decreases. I therefore find that this manuscript provides a clear and interesting analysis to answer some questions raised by the complex dynamics of supercooled liquids.

Weaknesses

1) My only downside concerns the structural analysis around the eigenmode cores. The analysis is conducted from a description of the polyhedra obtained by a Voronoï construction. I found that the latter only shows fairly small differences. It would have been interesting to carry out other structural analysis such as, for instance, the one carried out in [H. Tong and H. Tanaka Phys. Rev. X 8, 011041 (2018)] which has shown an excellent correlation with the dynamics of supercooled liquids.

Report

This article examines numerically the spatial structure of unstable vibrational modes in glass-forming liquids. It can be seen as an extension of a first study [D. Coslovich, A. Ninarello and L. Berthier, SciPost Phys. 7, 077 (2019)] in which it has been shown that, for a variety of fragile liquids, the fraction of delocalized unstable modes goes to zero as temperature is decreased, and equal to zero below to the mode coupling temperature T_MCT. Here, the spatial structure of the unstable modes and the relations with the structure is studied in more depth.

This work uses a three-dimensional ternary system with system sizes ranging from 1000 to 3000 atoms. Sampling of dynamical configuration of liquids is performed for different temperatures using a new SWAP Monte Carlo algorithm to thermodynamically equilibrate systems below the mode coupling temperature. The eigenmodes of vibration are calculated from the diagonalization of the Hessian matrix. The analyzed configurations are obtained from the minimization of the sum of the squares of the forces in order to find the stationary points of the potential energy landscape. The minimization procedure is followed by an eigenvector following search method in order to obtain properly (i.e. order of saddle points) converged stationary points. Several indicators are studied: the participation ratio, to look at the degree of localization of modes, and the average spatial profiles of displacement and energy of each mode. The local structure is also analyzed (radial distribution functions and Voronoï analysis) and comparisons are carried out between the soft quasilocalized modes of the minima and the saddle points.

The manuscript is well written with clear figures and all technical details are well explained. I find this article to provide interesting insights into the unstable modes in real space of finite dimensional liquids. It reinforces the conclusion of [D. Coslovich, A. Ninarello and L. Berthier, SciPost Phys. 7, 077 (2019)] by showing that the saddle modes localize as the temperature drops and shows that the delocalized modes disappear below T_MCT. It is moreover shown that the localized unstable modes converge towards the quasi-localized soft modes commonly observed for minima as the temperature of the liquid decreases. I therefore find that this manuscript provides a clear and interesting analysis to answer some questions raised by the complex dynamics of supercooled liquids and deserves to be published in SciPost.

My only downside concerns the structural analysis around the eigenmode cores. The analysis is conducted from a description of the polyhedra obtained by a Voronoï construction. I found that the latter only shows fairly small differences. It would have been interesting to carry out other structural analysis such as, for instance, the one carried out in [H. Tong and H. Tanaka Phys. Rev. X 8, 011041 (2018)] which has shown an excellent correlation with the dynamics of supercooled liquids.

Beside this question, I think the authors should also answer the following remarks and questions:

  • In equation 6, why use the median and not the mean. Does the presence of outliers question the analysis?

  • I find the paragraph starting with ”To briefly summarize the results of Ref. [26] using our data, we show the scatter plots of the participation ratio in Fig. 1. We show the results at (a) T = 0.45, (b) T = 0.35, (c) T = 0.32, (d) T = 0.30, and (e) T = 0.28. ” is not very clear and not summarizing well reference [26]. The authors should explicitly explain that, above lamdba_e, the participation ratio tends towards zero as the temperature decreases. Additionally, I found the introduction and definition of mobility edge clearer in [26].

  • The authors write: ”Nevertheless, we can also see a difference between the delocalized unstable modes and the QLVs” I find this not very clear in Figure 4.

Requested changes

1 -In section II, part A, concerning the sample preparation, the authors explain that they have improved the convergence of the sampling method. Could the authors provide arguments to explain that the states sampled in this way are statistically representative of equilibrium dynamics.

2- Although this is a subject already treated in [D. Coslovich, A. Ninarello and L. Berthier, SciPost Phys. 7, 077 (2019)], could the authors specify what is the fraction of the delocalized unstable modes as a function of temperature?

3- In section II.B.3, concerning the fractal dimension, it took me a while to understand that the authors were considering the integer part of the participation ratio. I must admit that I was not familiar with these signs. This is explained further: “In each box, we only show the dP (λ α) e particles having the largest norms, where [. . . ] denotes the integer part. " I suggest putting this explanation before at the first occurrence in the text.

4- The authors introduce “e^⊥_e α,ij” as the transverse relative displacement. I think the authors have to take the square root for it to be a displacement.

5- Still regarding formulation, the authors introduce the energy profile, but shouldn't we rather speak here of the energy variation profile?

6- Table I shows the absolute values of the eigenvalues. I find this to be a bit misleading. Also, I didn't understand whether it was an interval or not.

7- The scaling laws discussed in figures 3 and 4 are based (this is also the case in figure 12) on observations covering less than half a decade on the abscissa due to the smallness of systems (as discussed in the article). This aspect of things, giving a more qualitative picture, should be discussed more in the article.

8- ”We can clearly see the cores of the localized modes modes in Fig. 2 (a) and (b), while it is difficult to identify similar cores in the delocalized modes, at least by visual inspection. ” There is a typo in the repetition of the word “modes”.

9- For more clarity, the authors should add “saddle” and “minima” in the legend of figure 8, as in figure 7.

10- It is written : ”We found that Voronoi signatures in saddles and minima have markedly different statistics at T = 0.45.” I do not agree. I don't find Voroinoi's signature statistics to be different so markedly. Or, the authors should explain why this can be seen as a significant deviation.

11- It is explained that: “We restrict the calculation of the radial distribution function g (r) to central particles of species 3 that form the cores of the unstable modes, i.e., the particles whose index is i e †.” Could the authors justify this choice further? (I read the footnote).

12- It is written : “Overall, the structure around core particles is almost featureless and resembles the one of the fluid at higher temperature” To show this, it would have been useful to compare these radial distribution functions with that of a liquid at high temperature. Here the reader is forced to compare with Figure 7 and I find it a pity for such an interesting result.

  • validity: high
  • significance: good
  • originality: good
  • clarity: high
  • formatting: excellent
  • grammar: excellent

Author:  Masanari Shimada  on 2020-11-24  [id 1060]

(in reply to Report 2 on 2020-10-18)

This article examines numerically the spatial structure of unstable vibrational modes in glass-forming liquids. ... The local structure is also analyzed (radial distribution functions and Voronoi analysis) and comparisons are carried out between the soft quasilocalized modes of the minima and the saddle points.

The manuscript is well written with clear figures and all technical details are well explained. I find this article to provide interesting insights into the unstable modes in real space of finite dimensional liquids. It reinforces the conclusion of [D. Coslovich, A. Ninarello and L. Berthier, SciPost Phys. 7, 077 (2019)] by showing that the saddle modes localize as the temperature drops and shows that the delocalized modes disappear below $T_{\mathrm{MCT}}$. It is moreover shown that the localized unstable modes converge towards the quasi-localized soft modes commonly observed for minima as the temperature of the liquid decreases. I therefore find that this manuscript provides a clear and interesting analysis to answer some questions raised by the complex dynamics of supercooled liquids and deserves to be published in SciPost.

We are grateful to the referee for reading our manuscript carefully and appreciating the results. Below, we answered all the questions and comments raised by the referee.

My only downside concerns the structural analysis around the eigenmode cores. The analysis is conducted from a description of the polyhedra obtained by a Voronoi construction. I found that the latter only shows fairly small differences. It would have been interesting to carry out other structural analysis such as, for instance, the one carried out in [H. Tong and H. Tanaka Phys. Rev. X 8, 011041 (2018)] which has shown an excellent correlation with the dynamics of supercooled liquids. Beside this question, I think the authors should also answer the following remarks and questions:

To address the point raised by the referee, we have calculated the Tong/Tanaka structural order parameter $\Theta$ for our ternary model. This order parameter measures the average local deviation from close packing of neighboring particles, and has been shown to correlate quite strongly with the local dynamics in some models of supercooled liquids.

We carried out this analysis at $T=0.28$. We found that $\Theta$ is significantly larger for particles belonging to the unstable modes' cores than for the bulk, as demonstrated by the shift in corresponding probability density $p(\theta)$. Since $\Theta$ measures deviations from ideal close packing, we conclude that the cores are associated to less compact local environments. We also calculated the "precision" with which $\Theta$ can predict the cores. We found that the overlap between the particles with the largest $\Theta$-values and those belonging to the cores is higher than expected from randomly chosen particles, but still relatively low. This problem may be partly attributed to the presence of spurious second shell neighbors in the Voronoi tessellation.

In conclusion, this extended analysis reveals that, in our model, unstable cores are localized around steric defects associated to "loose" local packings. However, it remains difficult to predict which particles will form the cores.

In the revised manuscript, we have added a new subsection II.C.2 to introduce the $\Theta$ order parameter and we have added a new Fig.11 to present the main results, which we discuss at the end of Sec. II C, along with a brief comment in the Conclusions.

In equation 6, why use the median and not the mean. Does the presence of outliers question the analysis?

We followed the convention of Ref. [29], which first introduced the decay profile of the quasi-localized modes in stable configurations. Even if we used the mean instead of the median, the effect of the outliers would be negligible, because it is suppressed by the spherical average.

I find the paragraph starting with "To briefly summarize the results of Ref. [26] using our data, we show the scatter plots of the participation ratio in Fig. 1. We show the results at (a) T = 0.45, (b) T = 0.35, (c) T = 0.32, (d) T = 0.30, and (e) T = 0.28. " is not very clear and not summarizing well reference [26]. The authors should explicitly explain that, above $\lambda_e$, the participation ratio tends towards zero as the temperature decreases. Additionally, I found the introduction and definition of mobility edge clearer in [26].

We thank the referee for this useful comment. We revised the paragraph so as to follow more closely Ref. [26].

The authors write: "Nevertheless, we can also see a difference between the delocalized unstable modes and the QLVs" I find this not very clear in Figure 4.

We agree that the difference between the delocalized unstable modes and the QLVs is subtle. We have changed our discussion as follows:

"Nevertheless, we can also see a small difference between the delocalized unstable modes and the QLVs, particularly at $T\geq0.35$: the latter saturate at large distances. Although subtle, this difference means that the particles contributing to the QLVs are denser and more compact than those contributing to the delocalized modes."

1- In section II, part A, concerning the sample preparation, the authors explain that they have improved the convergence of the sampling method. Could the authors provide arguments to explain that the states sampled in this way are statistically representative of equilibrium dynamics.

These improvements refer to the "optimization" algorithm, not to the "sampling" algorithm. We have checked the manuscript and we confirm that this is clear from the text, even in our initial submission.

2- Although this is a subject already treated in [D. Coslovich, A. Ninarello and L. Berthier, SciPost Phys. 7, 077 (2019)], could the authors specify what is the fraction of the delocalized unstable modes as a function of temperature?

We have added these fractions to Table I.

3- In section II.B.3, concerning the fractal dimension, it took me a while to understand that the authors were considering the integer part of the participation ratio. I must admit that I was not familiar with these signs. This is explained further: “In each box, we only show the $\lceil P(\lambda_\alpha)\rceil$ particles having the largest norms, where [. . . ] denotes the integer part. " I suggest putting this explanation before at the first occurrence in the text.

We thank the referee for this useful comment. We have added an explanation in Section II.B.3.

4- The authors introduce $e^\perp_e \alpha,ij$ as the transverse relative displacement. I think the authors have to take the square root for it to be a displacement.

We have changed Sec. IIB4 in the revised manuscript based on this comment.

5- Still regarding formulation, the authors introduce the energy profile, but shouldn't we rather speak here of the energy variation profile?

If we correctly understand this comment, the energy profile in the manuscript actually measures the variation of the energy. It measures the vibrational energy of the system when a given mode is excited, and we do not include the total potential energy of saddles or minima.

In the revised manuscript, we pointed this out in Sec. IIB4 as follows: This measures the vibrational energy of the system when a given mode is excited, and we do not include the total potential energy of saddles or minima.

6- Table I shows the absolute values of the eigenvalues. I find this to be a bit misleading. Also, I didn't understand whether it was an interval or not.

We have changed the label of the table from $|\lambda|$ to $-\lambda$. These values are the central values (arithmetic averages) of the bins used to define the average $\langle \bullet \rangle_\lambda$ in Eq. (7), and the bin width is fixed at $\Delta \lambda=2.5$ throughout the manuscript.

7- The scaling laws discussed in figures 3 and 4 are based (this is also the case in figure 12) on observations covering less than half a decade on the abscissa due to the smallness of systems (as discussed in the article). This aspect of things, giving a more qualitative picture, should be discussed more in the article.

We agree with the referee that our system sizes are not large enough to confirm the precise asymptotic behavior of the profiles. Indeed, we already warned the reader about this in several occasions. We think it is already clear that our focus is more on the qualitative differences between the profiles of localized and delocalized modes, rather than on the quantitative determination of the scaling exponents.

8- "We can clearly see the cores of the localized modes modes in Fig. 2 (a) and (b), while it is difficult to identify similar cores in the delocalized modes, at least by visual inspection. " There is a typo in the repetition of the word “modes”.

We have fixed it, thanks.

9- For more clarity, the authors should add “saddle” and “minima” in the legend of figure 8, as in figure 7.

We have added the labels in Fig. 7.

10- It is written : "We found that Voronoi signatures in saddles and minima have markedly different statistics at $T = 0.45$." I do not agree. I don't find Voroinoi's signature statistics to be different so markedly. Or, the authors should explain why this can be seen as a significant deviation.

We have removed "markedly".

11- It is explained that: "We restrict the calculation of the radial distribution function g (r) to central particles of species 3 that form the cores of the unstable modes, i.e., the particles whose index is i e †." Could the authors justify this choice further? (I read the footnote).

Particles of different species have, on average, different local environments. To pinpoint the structural features of the core particles, we therefore restrict the calculation of the $g(r)$ to central particles of a given species. We consider only particles of species 3, because the vast majority of the core particles of these localized unstable modes belong to this species.

We have clarified this point in the revised manuscript and moved the corresponding footnote to the main text.

12- It is written: "Overall, the structure around core particles is almost featureless and resembles the one of the fluid at higher temperature" To show this, it would have been useful to compare these radial distribution functions with that of a liquid at high temperature. Here the reader is forced to compare with Figure 7 and I find it a pity for such an interesting result.

We checked the equilibrium $g(r)$ for central particles of species 3 and we found that one would have to consider temperatures much higher than 0.45 (the highest in this study) to observe a first minimum as flat as one of the core particles. Note, however, that the first peak for core particles remains higher than the equilibrium one.

To clarify these point and to emphasize that we do not make a quantitative comparison here, we changed the above sentence as follows:

"Overall, the structure around core particles is almost featureless and qualitatively resembles the one of the fluid at higher temperature. In particular, only at temperatures much higher than $T=0.45$ the equilibrium $g(r)$ would display a first minimum as flat as the one found for core particles."

---

## Round 1 · Referee Report · Anonymous (Referee 3) · 2020-10-19

Strengths

1) The authors throughly examine the spatial structure of unstable modes and provide clear differences between the unstable and stable modes.

2) The authors were able to study modes below close enough to the mode-coupling temperature to see a change of the modes.

3) The authors tested very clear hypothesis, and thus there are clear conclusions.

Weaknesses

1) Some of the specifics were not well defined in the paper. Specifically how the mode-coupling temperature was obtained and it was unclear how the authors defined the mobility edge.

2) The systems may be too small to properly see the full real space decay of the modes.

Report

Shimada, Coslovich, Mizuno, and Ikeda examine the structure of unstable saddle modes in supercooled liquids as a function of temperature, and compared these saddle modes with the stable modes. Since there were able to modify a previous algorithm to more efficiently find these unstable modes, they were able to examine the modes in more depth and for larger systems than in previous studies. To examine the spatial extent it is very important to be able to study as large a system as possible to limit any system size effects. However, since the spatial decay extends over half the box length in some figures, e.g. Figure 5 a, b, and c, these systems may still be too small. The authors do indicate that the system is too small to examine some asymptotic results.

The authors opened up a new pathway for future research in studying the spatial structure of unstable and stable modes, and future work with different systems, bigger systems, and at lower temperatures will eventually follow. It also opens up avenues for more detailed examination of the mode structure. The introduction is somewhat long but detailed and provides the reader with more than enough introduction to the subject, and the conclusion is clear, concise, and includes directions for future work. The description of the methods along with the citations to the relevant literature makes it possible for a researcher familiar with the field to be able to reproduce the results for this system and other systems.

Requested changes

1) It it not clear in the text exactly how the mode coupling temperature was determined. While it is expected to be the same as what was done it Reference 26, it would be helpful to the reader to know specifically how the mode coupling temperature was determined for this work.

2) In the discussion of the mobility edge, the reader is left to assume that the mobility edge is defined the same as in Reference 26. A simple sentence stating that this is the case would be helpful.

3) Can the authors estimate the size of the system needed to observe the asymptotic scaling expected in Figure 3?

4) As a minor point, the introduction is very long and the reader can become a little overwhelmed. The authors may consider making the introduction more focused.

  • validity: top
  • significance: high
  • originality: good
  • clarity: high
  • formatting: good
  • grammar: excellent

Author:  Masanari Shimada  on 2020-11-24  [id 1061]

(in reply to Report 3 on 2020-10-19)

The authors opened up a new pathway for future research in studying the spatial structure of unstable and stable modes, and future work with different systems, bigger systems, and at lower temperatures will eventually follow. It also opens up avenues for more detailed examination of the mode structure. The introduction is somewhat long but detailed and provides the reader with more than enough introduction to the subject, and the conclusion is clear, concise, and includes directions for future work. The description of the methods along with the citations to the relevant literature makes it possible for a researcher familiar with the field to be able to reproduce the results for this system and other systems.

We thank the referee for appreciating our manuscript.

1) It it not clear in the text exactly how the mode coupling temperature was determined. While it is expected to be the same as what was done it Reference 26, it would be helpful to the reader to know specifically how the mode coupling temperature was determined for this work. \newline

2) In the discussion of the mobility edge, the reader is left to assume that the mobility edge is defined the same as in Reference 26. A simple sentence stating that this is the case would be helpful.

In the revised manuscript, we now mention that $T_{\mathrm{MCT}}$ was estimated from the power law fit performed in Ref.50 and give a precise definition of $\lambda_e$, which is indeed the same as in Ref.26.

3) Can the authors estimate the size of the system needed to observe the asymptotic scaling expected in Figure 3?

In a previous study on the quasi-localized modes (Ref. [29]), systems of $10^6$ particles were needed to establish the scaling. This corresponds to the box length of about $L\simeq60$. We have added this explanation in the revised manuscript as follows:

"In Ref. [29], systems of $10^6$ particles were needed to establish this scaling."

4) As a minor point, the introduction is very long and the reader can become a little overwhelmed. The authors may consider making the introduction more focused.

We agree that the introduction is perhaps a bit long, but we feel that it presents all the necessary concepts and background.

Nonetheless, we did our best to shorten a few passages.

---

## Round 2 · Referee Report · Anonymous (Referee 2) · 2020-11-24

Strengths

A thorough study on a topic not yet addressed.

Weaknesses

No major weakness in particular.

Report

The authors have done excellent work in addressing all referees' concerns and in producing a better-revised version of the manuscript with a new analysis of the structure at the unstable cores. I recommend the publication of the manuscript.

---

## Round 2 · Referee Report · Anonymous (Referee 1) · 2020-11-24

Strengths

  • Clear
  • Insightful
  • Detailed

Weaknesses

No major weaknesses were identified.

Report

The resubmission addresses all the points raised in the first round of peer-reviewing. It also provides further analysis and insight on the local structure of the saddles and of the unstable cores. I do recommend publication in the current form.

---

## Round 2 · Author Response

We would like to thank all referees for their detailed feedback, which has provided us with an opportunity to clarify our results and improve our manuscript. In the following, we present our responses to all the concerns raised by all referees. Furthermore, we have revised the manuscript accordingly. We feel that these revisions have greatly improved our manuscript, and we hope that it is now suitable for publication in SciPost.

---

## Round 2 · List of Changes

1. Shorten the introduction slightly (referee 2).
  2. Add a note in Sec. IIA saying that a few crystallized samples were removed from the analysis
  3. Add the definition of the mode-coupling temperature (referee 3).
  4. Add a more detailed description of convergence issues of the EF optimizations in Sec. IIA (referee 1).
  5. Add the definition of the ceiling function in Sec. IIB3 (referee 2).
  6. Extend the explanation of the energy profile in Sec. IIB4 (referee 2).
  7. Change the dashed lines in Fig. 1 to the solid lines (referee 1).
  8. Revise the definition and discussion of the mobility edge in Sec. IIIA based on Ref. [26] (referee 2 and 3).
  9. Add the fraction of delocalized modes in Tab. I (referee 2).
  10. Add an estimation of the system size needed to observe the asymptotic scaling in Fig. 3, see the footnote on page 7 (referee 3).
  11. Weaken the emphasis on the difference between unstable delocalized modes and the QLVs in Fig. 4 (referee 2).
  12. Weaken the emphasis on the difference between Voronoi signatures (referee 2).
  13. Add the calculation of the Tong-Tanaka Theta order parameter to characterize the local structure of the core particles (referee 2)

---

## Editorial Decision

published